# Anomalous Spectral Shift of o-Modes in Multilayer Photonic Structure Induced by Homeotropic–Homeoplanar Transition in Chiral–Nematic Defect Layer

Vladimir A. Gunyakov [1] , Vitaly S. Sutormin [1,2] , Ivan V. Timofeev [1,2,*] , Vasily F. Shabanov [1]
and Victor Ya. Zyryanov [1]

[1]  Kirensky Institute of Physics, Federal Research Center KSC SB RAS, 660036 Krasnoyarsk, Russia;
gun@iph.krasn.ru (V.A.G.); sutormin@iph.krasn.ru (V.S.S.); shabanov@ksc.krasn.ru (V.F.S.);
zyr@iph.krasn.ru (V.Y.Z.)

[2]  Institute of Engineering Physics and Radio Electronics, Siberian Federal University,
660041 Krasnoyarsk, Russia

*   Correspondence: tiv@iph.krasn.ru

**Abstract:** A chiral nematic is embedded between multilayer mirrors to obtain voltage-inducible polarized resonance spectra. Initially, the nematic director is uniformly oriented perpendicular to the mirrors' surfaces because the chiral nematic helix is completely untwisted due to the homeotropic boundary conditions specified by the adsorbed cations. Then, a voltage is applied to remove the layer of surface-active cations from the input mirror. The obtained twisted homeoplanar configuration has a helix pitch exceeding the layer's thickness. The twisting leads to the anomalous blue shift of the o-modes in the transmittance spectrum of the photonic structure. This blue shift can be effectively compensated by repulsion of spectral peaks as a result of mode coupling in the vicinity of the virtual avoided crossing point. The experimental results obtained are confirmed numerically using the $4 \times 4$ transfer matrix method and explained with the contribution of a geometric phase.

**Keywords:** liquid crystal; hybrid aligned cholesteric; ionic surfactant; multilayer photonic structure; geometric phase





## 1. Introduction

The optics of liquid crystals (LCs) is remarkable for the variety of interrelations between the observed physical phenomena [1]. In particular, there exists an interrelation between the so-called geometric phase, also referred to as a topological phase, and some phenomena in quantum, relativistic, and classical physics, for example, the Pancharatnam–Berry phase in polarization optics. According to the definition, the geometric phase is a phase that occurs when an oscillatory system traces a closed curve in the parameter space. A simple example is the rotation of a Foucault pendulum when it moves along the approximately spherical surface of the globe. The rotation of a pendulum is proportional to the curvature and area covered by the pendulum's trajectory according to the solid angle theorem. A similar phenomenon in optics was described by Rytov in 1938 and Pancharatnam in 1956 [2,3]. The phenomenon gained great popularity after Berry's works on quantum mechanics [4]. For the Foucault pendulum, the characteristic time of transfer around the globe (days) is much larger than the pendulum oscillation period (seconds). When the times become comparable, a nonadiabatic (diabatic) Aharonov–Anandan phase arises [5,6].

The geometric phase can be controlled independently of the total phase, and therefore, it is used for a nondispersive frequency shift in modulators, when the mechanical rotation frequency of a phase plate is added to the field frequency [7]. The wavefront can be arbitrarily designed with the so-called geometric phase optical elements [8]. A simple

example is a quarter-half-quarter (QHQ) device consisting of two quarter-wave phase plates (to transform the linear polarization to a circular one) and one half-wave phase plate (to change the circular polarization phase via mechanical rotation). This approach allows the explanation of such phenomena as the transformation of light beams with orbital momentum [9–15], the formation of a basis for quantum computing [16], and the principle of operation of various tunable photonic structures such as anisotropic diffraction gratings, holograms [17–20], and lenses [21].

An important factor for the geometric phase is the chirality of an anisotropic medium, where the mirror symmetry vanishes because the local optical axis smoothly changes its direction at different points of a medium in the plane perpendicular to the light beam propagation direction. Chiral molecules form chiral photonic structures via self-assembly, which occurs in many biological materials. In this context, chiral liquid crystals are also nature-like materials [22]. Special attention should be paid to the nonadiabatic phase, which arises when the adiabaticity is violated, both in difficult-to-implement quantum physics tasks [23] and in experimentally accessible problems of classical optics of liquid crystals [24], as well as in photonic lattices [25]. Thus, a nontrivial mathematical relation between the seemingly different optical phenomena can be traced [1].

An experimentally implementable chiral defect can be a hybrid aligned cholesteric (HAC). This structure is formed when a cholesteric is placed in a cell where it has a uniform planar surface anchoring at one substrate and a homeotropic one at another [26–31]. A simultaneous tilt and twist of the director along the LC layer normal is observed in the HAC. Depending on the ratio between the LC layer thickness $d$ and cholesteric helix pitch $p$, a homogeneous or modulated HAC structure can be realized [26,28,29]. A modulated HAC acts on the transmitted light as a phase diffraction grating, which can be controlled with various external factors [30,31]. It should be noted that a HAC structure can also be observed at the modification of the boundary conditions in the LC cell. In [32], the formation of a uniform or modulated HAC was observed at the electrically controlled ionic modification of the surface anchoring from homeotropic to planar on one of the LC cell substrates.

In this work, the geometric-phase-induced anomalous spectral shift of eigenmodes is experimentally demonstrated in a multilayer photonic structure containing a defect in the form of a chiral nematic layer with the homeoplanar surface anchoring, which in fact, is a HAC. Under the conditions of the structural transition from a homeotropic director configuration to a homeoplanar one in a chiral nematic defect layer induced by the ionic modification of the surface anchoring, features of the components of the polarized transmittance spectrum of the structure were examined in the short-wavelength range, where the strong polarization mixing of modes was manifested. The experimental data were compared with the results of the numerical simulation using the $4 \times 4$ transfer matrix method.

## 2. Experimental

A photonic structure (PS) consists of two multilayer mirrors and includes chiral nematic (CN) as a defect layer: Sub/H(LH)$^5$ITO–CN–ITO(HL)$^5$H/Sub (Figure 1a), where L (SiO$_2$) and H (ZrO$_2$) are the optically isotropic oxide layers with the low (nL = 1.45) and high (nH = 2.05) refractive indices, respectively. The number of LH and HL bilayers to the left and right of the CN layer is five. Quartz glass substrates (Sub) were used. The investigated structure is hereinafter referred to as PS/CN. At thicknesses of ~(82 ± 5) nm for the silicon oxide layer and ~(63 ± 5) nm for the zirconium oxide layer, a photonic band gap (PBG) formed in the transmittance spectrum in the wavelength range of 425–625 nm. The violation of the periodicity of the structure led to the appearance of transmittance peaks in the band gap that corresponded to the optical modes localized on the defect layer.

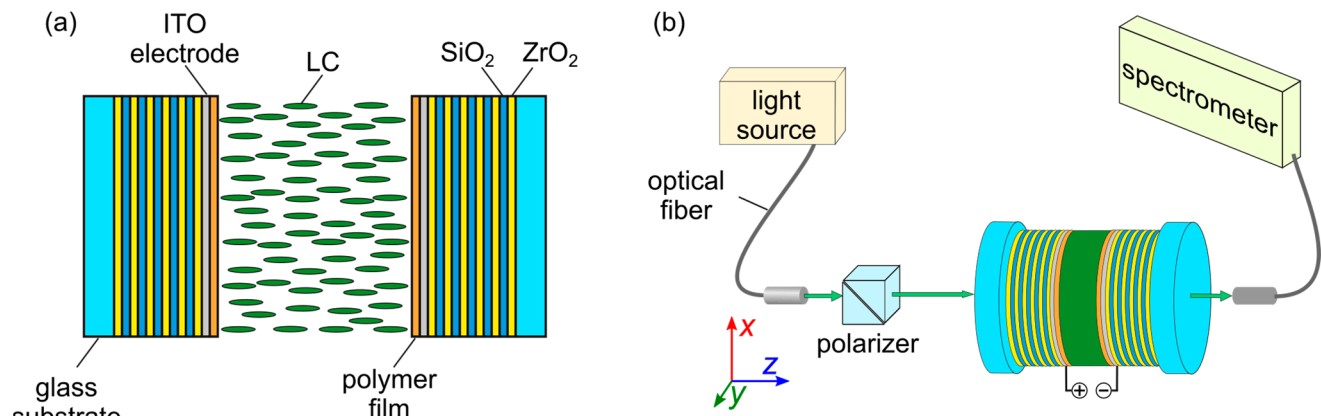

**Figure 1.** (**a**) PS/CN structure in the initial state (U = 0 V): The CN helix is untwisted, and the LC molecules are aligned normally to the mirror surfaces. (**b**) Scheme of the experimental setup for recording the polarized transmittance spectra.

Transparent ITO electrodes with a thickness of ~120 nm deposited onto the mirrors were used for the application of the electric field to the LC layer. The electrodes were coated with polyvinyl alcohol (PVA) (Sigma Aldrich) polymer films plasticized with a glycerol compound (Gl) in a weight ratio of PVA/Gl = 1:0.477 to form alignment layers for LC. The polymer films were deposited via spin coating, and the easy orientation axis was specified via unidirectional rubbing of the polymer surface. The defect layer thickness was $d = 7.22$ μm. The investigated nematic LC was 4-pentyl-4′-cyanobiphenyl (5CB) (Belarusian State Technological University) doped with a chiral additive cholesteryl acetate (ChA) (Sigma Aldrich) in a weight ratio of 5CB/ChA = 1:0.008. The cationic surfactant cetyltrimethylammonium bromide (CTAB) (Sigma Aldrich) was preliminarily added to the nematic in a weight ratio of 5CB/CTAB = 1:0.007. CTAB, dissolved in 5CB, dissociated into the positively charged surface-active ions $CTA^+$ and negative ions $Br^-$. The CTAB additive made it possible to obtain the orientational transition in the LC layer induced by the electrically controlled ionic modification of the surface anchoring [32,33]. In the mixture used, the CN helix pitch was $p = 20.7$ μm, so the ratio $d/p$ between the defect layer thickness and the helix pitch was 0.35. The polarized transmittance spectra of the photonic structure in the homeotropic–homeoplanar structural transition regime were experimentally studied on a setup based on an Ocean Optics HR4000 spectrometer equipped with fiber optics (Figure 1b). A Glan prism was used as a polarizer. The temperature-controlled sample ($t = 23$ °C) was set in such a way that the directions $R_{1,2}$ of rubbing PVA surfaces on the mirrors were parallel to the $x$-axis of the laboratory coordinate system ($x$, $y$, and $z$). A dc electric field in the form of a rectangular pulse with a length of 10 s and an amplitude of several volts from an AKTAKOM AHP-3122 generator was applied to the sample. Applied voltage $U$ was detected with a Keysight Technologies 34465A digit multimeter.

Figure 2a,b show a scheme of the homeotropic–homeoplanar transition in the CN layer under the action of a dc electric field pulse. In the initial state, the surface-active ions at a sufficient concentration in LC screen the planar aligning effect of the polymer coatings and specify the normal boundary conditions for LC molecules. As mentioned above, in the test sample, we have $d/p < 1$; therefore, the CN helix is completely untwisted, and the homeotropic director configuration is realized (Figure 2a). The dc electric field applied to the PS/CN cell forces the surface-active $CTA^+$ ions to leave the mirror with electrode–anode. As a result, the planar boundary conditions specific to the polymer film are formed on this mirror: the local director in the LC surface layer is parallel to the rubbing direction $n_{in} \parallel R_1$. At the same time, the boundary conditions on the mirror with electrode–cathode remain invariable. Thus, on the one hand, the asymmetric planar–homeotropic boundary conditions are formed in the nematic layer, which are characteristic of the homeoplanar director configuration [24] and, on the other hand, the ionic processes

within the 5CB layer release the twisting force of the ChA chiral additive, which results in the homeotropic–homeoplanar transition (Figure 2a,b).

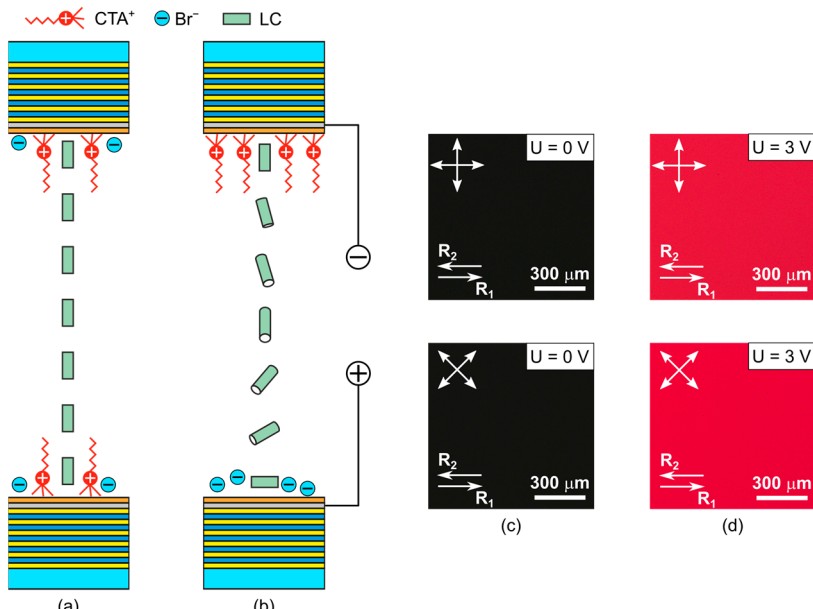

**Figure 2.** (**a**,**b**) Scheme of the homeotropic–homeoplanar structural transition in the CN layer and (**c**,**d**) microphotographs of the PS/CN cell textures taken in crossed polarizers. Textures (**c**,**d**) correspond to homeotropic (**a**) and twisted homeoplanar (**b**) director configurations, respectively. $R_1$ and $R_2$ are the PVA rubbing directions. Double arrows indicate the polarizer and analyzer directions.

This transition changes the optical properties of the PS/CN cell, which can be seen in the microphotographs of the textures of the investigated sample (Figure 2c,d) taken in crossed polarizers. In the initial state, the observed PS/CN cell texture is a uniform, dark region (Figure 2c), regardless of the angle of sample rotation on a microscope stage, which indicates the formation of a homeotropic director configuration. This texture remains unchanged up to a certain voltage threshold of $U = 2.75$ V, at which the light transmittance of the sample drastically increases. In the voltage range of $2.75 \leq U \leq 3.25$ V, the texture is a uniform bright area, and a dark state is not observed during the sample rotation on the microscope stage, indicating the formation of a twisted homeoplanar director configuration. Figure 2d shows, in particular, the textures of the sample oriented at angles of 0° and 45° between the rubbing direction $R_1$ on the input mirror and the polarizer. It should be noted that these ionic processes occur relatively fast. For example, the time of response to an electric pulse in an LC cell with similar parameters is tenths of a second [32]. This implies the rapid formation of a stable, twisted homeoplanar configuration and its stability during the action of an electric pulse, which is important for recording the spectra of the optical modes of the investigated structure. In addition, it is noteworthy that the multiple twisting of the LC followed by its relaxation did not affect the reproducibility of the textures and spectra of the photonic structure. After the end of the electric pulse action, the structure always returns to its initial state. The electrohydrodynamic instabilities arise in the LC layer at a larger value of the dc electric field [32].

## 3. Results and Discussion

The transmittance spectra of the PS/CN cell were recorded both in the absence of an electric field and during the action of an electric pulse. In the latter case, the field amplitude in the pulse changed with a step of 0.25 V in the voltage range of $0 \leq U \leq 3.25$ V. In the case of homeotropic director configuration, linearly polarized light of any azimuth, for each pass inside a resonator, will maintain its polarization state since the light wave vector E remains perpendicular to the director: n ∥ z. In particular, a

linear polarization along the *x*-axis within the untwisted CN does not leave the *xz* plane (Figure 1b). Therefore, the transmittance spectrum of the structure consists of well-resolved single peaks corresponding to the cavity *o*-modes, and the series of these modes within the PBG is, in fact, the polarized $T_\perp$ component. Below, the $T_\perp$ component is presented in the figures in the form of a spectrum of the initial state of the investigated structure as a background to illustrate the transformations of the modes during the transition.

Figures 3 and 4 show the polarized transmittance spectra that were obtained in the experiment and calculated using the $4 \times 4$ transfer matrix method for the investigated photonic structure obtained at the longitudinal (E $\parallel$ $n_{in}$, Figure 3) and transverse (E $\perp$ $n_{in}$, Figure 4) orientation of the polarizer relative to the field-induced direction of the local director $n_{in}$ on the input mirror. The o-modes in Figure 4 are well-resolved in an experiment that is crucial for the detection of tiny shift values less than 1 nm. The background is the spectrum of the $T_\perp$ component, the resonance peaks of which remain in their stable positions up to a threshold voltage of $U = 2.75$ V. On the amplification of applied voltage, two distinct positions of the peaks are resolved as shown in Figure 4a. The peculiarity of ion-surfactant reorientation is that it occurs sharply at critical surface concentration of the adsorbed cations. The first set of spectral peaks vanishes, then another set appears. Therefore, the shift of the spectra is a jump, and a smooth shift is observed neither under nor above the threshold voltage $U = 2.75$ V. It can be seen that, above the threshold, the mixed $T_e$ and $T_o$ components are excited in the cavity, the spectra of which contain—along with resonance peaks corresponding to the main polarization of the mode—satellite peaks of the lower-intensity orthogonal polarization. This feature especially appears in the short-wavelength spectral range (Figures 3 and 4). The presence of satellite peaks is typical for chiral photonic structures and indicates a violation of the waveguide mode of propagation of linearly polarized waves [34]. This means that in the twisted homeoplanar CN layer, the elliptically polarized twist extraordinary (*te*) and twist ordinary (*to*) waves circulate, which, reflecting from the mirrors, are coupled with each other and thereby generate the cavity eigenmode [35]. The polarization state and, consequently, the type (extraordinary *re* or ordinary *ro*) of cavity mode depend on the amplitude ratio between the partial twist waves forming this mode. Despite the variable ellipticity of the cavity modes in a chiral medium, their polarization on the mirrors is linear. If the *re* and *ro* cavity modes tend to cross in the vicinity of the point corresponding to the maximum possible coupling of the partial *te* and *to* waves, then the avoided crossing phenomenon should be expected [36]. The essence of this phenomenon is that the linear polarization of the approaching *re* and *ro* modes does not coincide with the direction parallel (*re*) or orthogonal (*ro*) to the local director of the layer adjacent to the mirror. The angle of divergence between the mode polarization and the director determines the amplitude of the satellite peak in the spectrum. The maximum discrepancy $\pm 45°$ leads to the equalization of the amplitudes of the main peak and the accompanying satellite peak. The modes corresponding to them repel each other apart relative to the avoided crossing point while exchanging the polarization state *re* $\leftrightarrow$ *ro*. In particular, the role of the avoided crossing point is played by the wavelength $\lambda_c$, at which the partial cavity modes would intersect in the absence of coupling on the mirrors [36]. The investigated PS/CN structure is not an exception in a series of chiral photonic crystals. The polarization mode mixing in the vicinity of a wavelength of $\lambda_c = 463$ nm shows that it can be accepted as a virtual avoided crossing point (Figures 3 and 4). Indeed, as this point is approached, both from the left—taking into account the background—and from the right, the amplitudes of the satellite peaks gradually increase, tending to equalize with the main polarization peaks. In addition, when passing through this point, each satellite changes positions with the accompanying main peak, which leads to the formation of a symmetric spectral profile for both components at these frequencies.

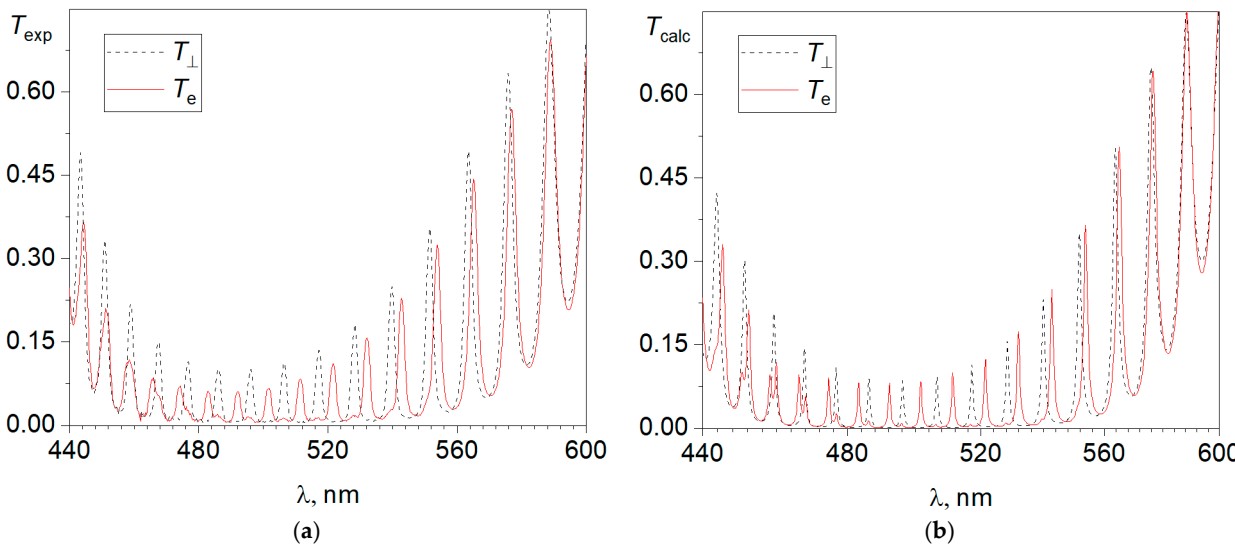

**Figure 3.** (**a**) Experimental transmittance spectra and (**b**) spectra calculated with the 4 × 4 transfer matrix method for the PS/CN cell (solid lines) obtained at a longitudinal orientation of the polarizer (E ∥ n$_{in}$) and a voltage pulse amplitude of $U$ = 3 V. The dashed lines show the spectrum of the $T_\perp$ component.

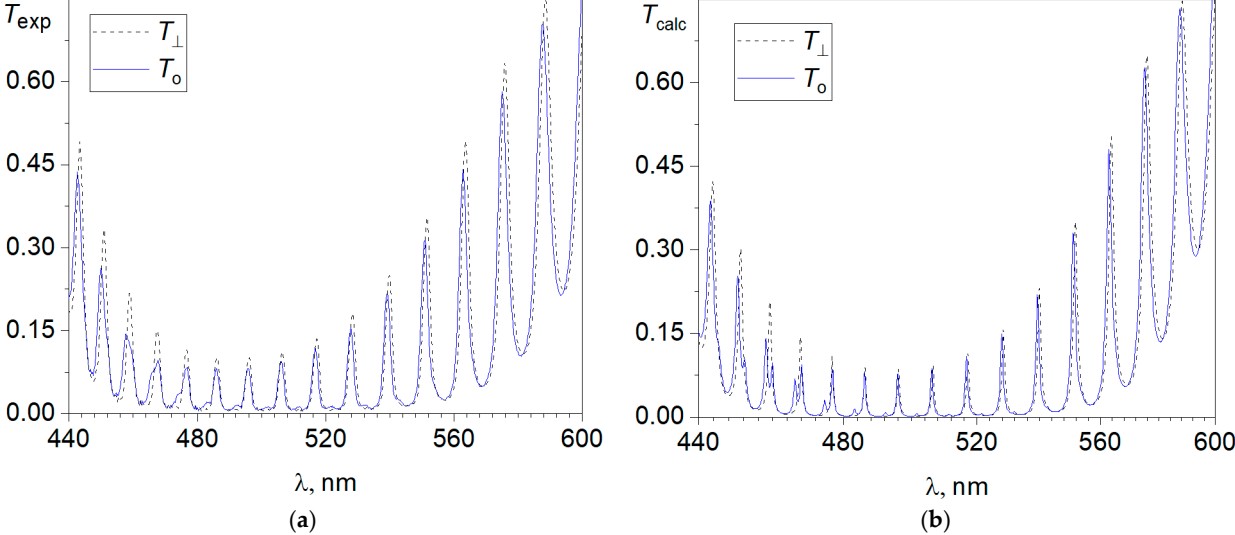

**Figure 4.** (**a**) Experimental transmittance spectra and (**b**) spectra calculated with the 4 × 4 transfer matrix method for the PS/CN cell (solid lines) obtained at a transverse orientation of the polarizer (E ⊥ n$_{in}$) and a voltage pulse amplitude of $U$ = 3 V. The dashed lines show the spectrum of the $T_\perp$ component.

Above a wavelength of 500 nm in the long-wavelength spectral range, the $T_e$ and $T_o$ components have several single peaks (Figures 3 and 4). The modes corresponding to these peaks are polarized parallel (the *re* modes) or perpendicular (the *ro* modes) to the director n$_{in}$ on the input mirror. At the same time, the linear polarization of these modes at the output from the photonic structure turned out to be rotated by an angle of 87° relative to these directions, which corresponds to the calculated total angle of director twisting in the CN. However, this is just a simulation of the Mauguin waveguide regime, since the modes in the bulk of the LC, as shown with the calculation, have elliptical polarization. The comparison of the spectra presented in Figures 3 and 4 also shows that each component feels its own refractive index. This appears in the difference between the free spectral ranges of the resonator: in the presented spectral range, there are 16 resonance peaks of the $T_e$

component and 15 resonance peaks of the $T_o$ component. In the initial state, the LC medium is homogeneous and for any azimuth of linearly polarized light, the refractive index remains constant, $n_o = n_\perp$. As a result of the homeotropic–homeoplanar structural transition, a certain value of the effective refractive index of the medium is realized for the extraordinary wave: $< n_e > = (1/d) \int_0^d n(z)dz$. Thus, above the threshold voltage, the ionic modification of the boundary conditions leads to the transformation of the symmetric photonic structure into an asymmetric microcavity with a specific distribution of the refractive index in the LC layer. The end of the action of the electric pulse returns the LC layer to its initial state with the homeotropic configuration. In this case, the *re* and *ro* modes collapse and take their initial spectral positions.

The transmittance spectra of the investigated photonic structure were simulated using the Berreman $4 \times 4$ transfer matrix [37] with regard to the decay of defect modes [38] and the dispersion properties of the materials that form the sample. The twisted homeoplanar LC configuration was calculated by minimizing the energy of elastic deformations of the director field n(r) under asymmetric boundary conditions. The LC surface anchoring one of the mirrors was planar and was homeotropic on the other. The parameters used in the calculation included 5CB splay, twist, and bend elastic constants of $K_{11} = 5.7$ pN, $K_{22} = 3.5$ pN, and $K_{33} = 7.7$ pN, respectively, as well as a defect layer thickness of $d = 7.22$ μm and a helix pitch of $p = 20.7$ μm. Figure 5 presents the distribution of the orientation angles $\theta(z)$ and $\varphi(z)$ of the twisted homeoplanar director configuration formed under the action of an electric pulse inside the defect layer of the photonic structure. It can be seen that the director is tilted and twisted simultaneously in the LC bulk. The total director rotation angle is $\varphi \cong 87°$. Using these angle distributions, the transmittance spectra $T_e(\lambda)$ and $T_o(\lambda)$ for the PC/LC structure were calculated. These spectra, on the one hand, are in good agreement with the profile of the experimental curves in the band gap and, on the other hand, give adequate spectral positions of the modes (Figures 3 and 4). In addition, the simulation of the spectra can significantly improve their resolution. This is especially important for the short-wavelength range, where the resonance peaks of the main polarization are accompanied by satellite peaks of the orthogonal polarization. Apart from the transmittance spectra, the transfer matrix method makes it possible to calculate the distribution of the light field intensity for any mode and thereby determine its number $m$, which corresponds to the number of antinodes of standing waves localized in the cavity. At the homeotropic director alignment n ∥ z, any potential pair of orthogonally polarized modes with the number $m$ is degenerate. When the LC director reorients, the medium becomes birefringent, and the degeneration is eliminated. It leads to the splitting of each *o*-mode into cavity modes *re* and *ro*, which form two independent series. The behavior of the *re* modes in the spectrum is affected by the multidirectional trends caused by a change in the effective refractive index $<n_e>$ [39]. In this case, all *re* modes are shifted to the long-wavelength spectral region, crossing the nearest *ro* modes, which are fixed at certain wavelengths. From the corresponding numbers $m$, it can be concluded that the spectral positions of the *re* modes established as a result of the homeotropic–homeoplanar structural transition in the CN also follow the indicated trends. In particular, the *re*-mode cavity with the number $m = 36$ at a wavelength of 588.4 nm is shifted relative to its initial position at a wavelength of 551.6 nm, i.e., by three free spectral ranges of the resonator (Figure 3). At the same time, almost all *ro* modes in the CN undergo an anomalous blue shift relative to their initial spectral positions (Figure 4). In full accordance with the theoretical predictions, the shift increases towards longer wavelengths [24]. In particular, the *ro* mode with $m_o = 39$ at a wavelength of 516.7 nm is shifted by 0.5 nm, while the edge *ro* mode with $m_o = 33$ at a wavelength of 587.3 nm is shifted by 1 nm. The mechanism responsible for the observed anomalous shift of the *ro* modes of the photonic structure is the contribution of the geometric phase to the total phase shift acquired by the wave over a round-trip propagation through the cavity [24]. The phase mechanism is universal since it acts on all *ro* modes within the PBG.

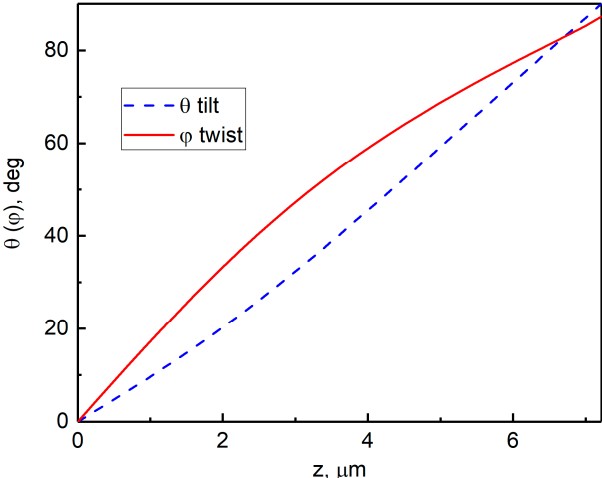

**Figure 5.** Calculated tilt angle $\theta(z)$ and twist angle $\varphi(z)$ of the CN director as functions of coordinate $z$. The coordinates $z = 0$ and $z = 7.22$ μm correspond to the mirrors with the planar and homeotropic surface anchoring, respectively. The nematic layer thickness is $d = 7.22$ μm. The helix pitch is $p = 20.7$ μm.

The particular interest is the spectral range in the vicinity of the virtual avoided crossing point $\lambda_c$, since, in this range, the phase mechanism of the anomalous blue shift of the *ro* modes is superimposed by a local, but stronger mode repulsion effect caused by the coupling of the *te* and *to* modes that form each cavity mode. Figure 6 shows the calculated spectra of the *re* and *ro* modes of the photonic structure, which appeared in the vicinity of the point $\lambda_c$ as a result of the structural transition. For comparison, the $T_\perp$ component indicating their initial spectral positions is shown. Let us consider the features of the mode shift in this range.

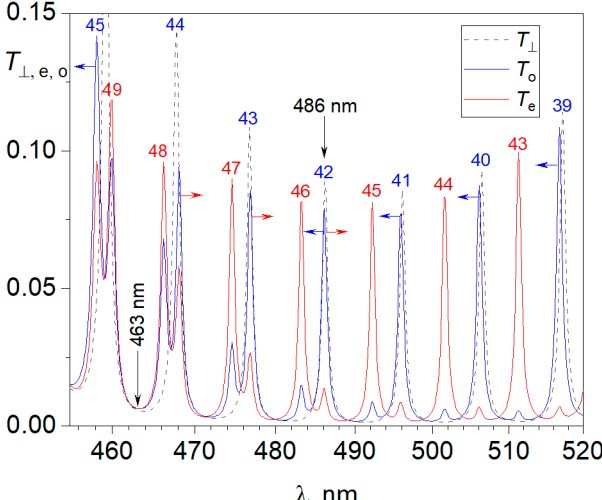

**Figure 6.** Competition between two counter mechanisms of shifting the *ro* modes of the photonic structure in the vicinity of the virtual avoided crossing point $\lambda_c = 463$ nm: repulsion effect (red arrows) and anomalous blue shift (blue arrows).

It can be seen from the figure that the *re* and *ro* modes in the spectrum are grouped into pairs. In the first pair, a blue shift of about 1.2 nm for the *ro* mode with $m_o = 45$ is more than twice the value of the same *ro* mode with $m_o = 39$ and even exceeds the maximum shift value of the edge *ro* mode, $m_o = 33$. The apparent paradox is explained by the fact that, to the left of the point $\lambda_c$, the *re* modes repel the *ro* modes to the short-wavelength region by the mode coupling. Thus, in this case, the repulsion effect and the phase mechanism

act codirectionally, thereby doubling the blue shift of the *ro* mode with $m_o = 45$. In the pairs to the right of the point $\lambda_c$, the *re* modes also noticeably affect the positions of the corresponding *ro* modes but repel the *ro* modes to the long-wavelength region, i.e., in the direction opposite to the blue shift. The mode coupling range is limited and, in addition, mode coupling noticeably weakens when moving away from the avoided crossing point. Indeed, despite the incursion of the geometric phase, the *ro* modes with $m_o = 44, 43$ are shifted to the red region of the spectrum relative to their initial positions: the near mode has a larger shift, and the far mode has a smaller shift. Therefore, it can be concluded that, in this wavelength range, the repulsion effect dominates due to the proximity of these modes to the point $\lambda_c$. Further, the *o*-mode with a number of $m_o = 42$, despite the significant transformation of the spatial distribution of the light field $o \rightarrow ro$ in a chiral medium, retains its spectral position at a wavelength of $\lambda = 486$ nm. This is evidence for the compensation of the counter-shift mechanisms that arise in the chiral photonic structure at the moment of the homeotropic–homeoplanar structural transition. Finally, for the *ro* modes with $m_o = 41$, 40, and 39, the phase shift mechanism dominates again.

In previous research, the geometric-phase-induced shift was described in detail for the twisted LC layer as a defect of the PhC structure that obtains helix symmetry of the LC director orientation field [24]. Later, similar dynamics was confirmed for the mirror-symmetric torsion reorientation of the LC layer. Finally, the homeoplanar orientation had no spatial symmetry signatures, nonetheless, the spectral shift was observed as well.

## 4. Conclusions

In this study, the experimental and numerical methods were used to examine the change in the polarization and spectral characteristics of the eigenmodes of the PS/CN multilayer photonic structure at the transition from a homeotropic director configuration to a homeoplanar one in the CN defect layer. The 4-pentyl-4′-cyanobiphenyl nematic liquid crystal doped with the cholesteryl acetate chiral additive and cetyltrimethylammonium bromide ionic surfactant was used as a defect. In the initial state, due to a ratio of $d/p$ = 0.35 between the layer thickness and the helix pitch, and the homeotropic boundary conditions specified by cations adsorbed on the mirrors, the chiral nematic helix was completely untwisted. Under the action of a dc electric field pulse, the ionic modification of surface anchoring occurred. In particular, one of the photonic structure mirrors became free from the layer of surface-active cations. On the contrary, the cation density on the other mirror increased. This resulted in the formation of a twisted homeoplanar director configuration. The optical textures of the PS/CN cell confirmed the formation of a CN helical structure during the action of the electric pulse. The twisted configuration of the director manifested itself as certain features of the $T_e(\lambda)$ and $T_o(\lambda)$ components of the polarized transmittance spectrum of the photonic structure. The first feature concerned the $T_o$ component and consisted of an anomalous blue shift of all the cavity *o*-modes within the band gap. The mechanism responsible for the observed *o*-mode shift was the occurrence of the geometric phase and its contribution to the total phase incursion acquired by the wave over a round-trip propagation through the cavity. The second feature affected both components and consisted in the fact that, in the vicinity of the virtual avoided crossing point $\lambda_c = 463$ nm, the phase mechanism of the anomalous shift of the *o*-modes was superimposed by the local mode repulsion effect caused by the coupling of the twisted *te* and *to* modes that formed each cavity mode. This effect acted on the *o*-modes in the opposite direction and made a stronger impact on their spectral position; however, at a sufficient distance from the point $\lambda_c$, it was reduced enough to be compensated for by the blue shift phase mechanism, as demonstrated at a wavelength of $\lambda = 486$ nm. It should be noted that the cardinal rearrangement of the light field in the cavity leads, as a rule, to the rearrangement of the spectral positions of the modes. However, in the PS/CN structure, the competition of differently directed mechanisms made it possible to implement the situation when both *o*-modes with the same number, but corresponding to the completely different spatial distributions of the light field, oscillated at the same frequency. The director field simulation

lacks two factors for non-ideal hybrid aligned cholesteric configuration. The first factor is the residual constant field, uncompensated by ions in the volume. The second factor is ionic current, which is the circulation of ions from substrate to substrate. Both factors make the experimental director field more homeotropic than the simulated one. The transmittance spectra of the PS/CN photonic structure were simulated using the $4 \times 4$ matrix transfer method. The experimental and calculated spectra are in good agreement with both the homeotropic and homeoplanar configurations of the CN director.

**Author Contributions:** Sample preparation, experimental setup, and measurement, V.A.G. and V.S.S.; software, I.V.T.; conceptualization and methodology, V.A.G., V.S.S. and I.V.T.; writing—review and editing, V.A.G., V.S.S., I.V.T., V.F.S. and V.Y.Z.; supervision, V.F.S. and V.Y.Z. All authors have read and agreed to the published version of the manuscript.

**Funding:** The work was carried out within the state assignment of the Federal Research Center KSC SB RAS.

**Institutional Review Board Statement:** Not applicable.

**Informed Consent Statement:** Not applicable.

**Data Availability Statement:** The data presented in this study are available upon reasonable request from the corresponding author.

**Conflicts of Interest:** The authors declare no conflict of interest.

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
