# Peer review of "Anomalous Spectral Shift of o-Modes in Multilayer Photonic Structure Induced by Homeotropic–Homeoplanar Transition in Chiral–Nematic Defect Layer"

_photonics, doi:10.3390/photonics10090959_

Round 1

Reviewer 1 Report

The manuscript by Gunyakov et al., reports an anomalous spectral shift of o-Modes in a photonic structure which is attributed to the occurrence of the geometric phase. The transmittance spectra of the photonic structure were simulated by the 4x4 matrix transfer method, the experimental and numerical data are in agreement with respect of the configurations of the nematic director.  

Overall, the manuscript is clear and well written and I consider it can be published as it is. I would like to see a future work analyzing the cases where the d/p ratio varies and its effect on the transmittance spectra and the occurrence, or not, of the geometric phase. 

Author Response

Please, see the attached pdf file

Reviewer 2 Report

In this manuscript, Authors designed a chiral nematic LC cell placed between multilayer mirrors and observed anomalous blue shift of the o-modes in the transmittance spectrum under an application of vertical DC field. They experimentally found the applied DC field to cause the anisotropic diffusion of CTA+ from dissolved CTAB in LCs toward the cathode and thus generate the twisted homeoplanar configuration. On the basis of spectral investigation and numerical calculation, they demonstrate the unique director configuration to be responsible for the anomalous spectral shift. 

Overall, the experiments are well designed and investigated logically. However, I have to admit that the current manuscript would not interest to general readers although Authors have explored intriguing optical phenomena in LCs. 

It is not an English issue. While the introduction and experiment sections are well organized and their contents are fully described step by step by which even non-expert can digest all contents, the result and discussion sections are not. Therefore, I am afraid that general readers would not be able to recognize the significance of this work.

A couple of examples are as follows:

1) If Authors show the spectrum below the threshold U, general readers would be easier to catch how the spectrum is modulated by the director reorientation.

2) More detailed explanation why the well-resolved single peaks in Figs. 3 & 4 corresponding to the cavity o-modes, would help general readers to follow Authors interpretations.

and so on..

I strongly suggest that Authors do not simply add references without any related explanation and assume your readers to be the experts on optics. I believe the revision of this manuscript in this way will dramatically improve its visibility. 

I would be happy to recommend the publication of this manuscript if Authors revise the manuscript in a satisfactory way.

Author Response

Please, see the attached pdf file

Reviewer 3 Report

This article is quite interesting in the point that the addition of a tiny amount of cationic surfactant greatly affects the property of LC cells. The manuscript is well-written and organized. The reviewer thinks that the manuscript should be accepted after minor revision.

Major issues

The authors should explain why they used CTAB as a cationic surfactant. What can be considered when an anionic surfactant was used?

Minor spell checks and typos should be corrected.

Line 64 thickness “d” and cholesteric helix pitch “p” should be italics.

Line 78: The multiple sign is missing.

Line 82 SiO2 and ZrO2: “2“should be lowercase.

Author Response

Please, see the attached file

Reviewer 4 Report

The work explores optical characteristics of a photonic structure consisting of two solid-state photonic structures and confined between them layer of liquid crystal. The layer of the liquid crystal plays the role of a defect in the photonic structure. Reorientation of the liquid crystal molecules from homeotropic on the two cell boundaries to homeotropic at one boundary and planar at the other boundary leads to modification of optical spectrum. Change of the orientation was performed using ionic additive to the liquid crystal and application of electric field. The authors of the manuscript are specialists in this area, they have published a number of articles in this direction. In the refereed manuscript a number of peculiarities of optical spectrum were experimentally observed, which were also calculated and interpreted. The manuscript is useful for readers of Photonics journal and can be published, however the referee has a number of remarks.

1. In point “1. Introduction” in page 1 (22 lines) the word combination “geometrical phase” is used 10 times. Meanwhile in the main part of the work (2. Experimental; 3. Results and Discussion) this word combination “geometrical phase” is used only 2 times. This gives an impression that in Introduction and in the main part speaks about different things. It would be possible to decrease the number of mentions of “geometrical phase” in the Introduction.

2. Modification of the spectra is performed with the use of electric field. Meanwhile the effect is the opposite with respect to observed without the ionic additive. It would be interesting for the readers whether it is possible to obtain again homeotropic orientation at larger value of the field since the dielectric anisotropy of 5CB is positive. Or it is not possible due to large conductivity of liquid crystal with the ionic additive and emergence of hydrodynamic instability?

3. In the section “2. Experimental” it is written that the layer thickness of liquid crystal is d=7.2 micron. In the caption of Figure 5 it is written that layer thickness is d=7.22 micron. Is there any sense in this difference?

4. The numbering of sections in the manuscript is as follows: 1, 2, 3, 5. Is it due to the authors mixing the numbers or there is section 4 which the authors forgot to include in the submitted manuscript?

In the referee’s opinion, after getting answers to these questions the manuscript can be accepted for publication.

-

Author Response

Please, see the attached pdf file

Round 2

Reviewer 2 Report

Authors have addressed my comments in a satisfactory way. I recommend the publication of this manuscript.